# Tetraploid Wheats: Valuable Source of Phytosterols and Phytostanols

**Othmane Merah** [1,2,*] [ID] **and Zephirin Mouloungui** [1] [ID]

[1] Laboratoire de Chimie Agro-industrielle, LCA, Université de Toulouse, INRA, 31030 Toulouse, France; zephirin.mouloungui@ensiacet.fr
[2] IUTA, Département Génie Biologique, Université Paul Sabatier, 32000 Auch, France
* Correspondence: othmane.merah@ensiacet.fr; Tel.: +33-534323523

**Abstract:** Phytosterols are known as healthy compounds obtained mainly from oilseed crops. Cereals were also studied for their sterols content. Few insights have been devoted to other tetraploid species than emmer and durum wheats. This work examined phytosterol and phytostanol content in seed of six tetraploid wheat species cultivated during two successive years under rainfed organic conditions in Auch (near Toulouse, France). Sterols (free and esterified sterols) were measured by gas-chromatography-flame ionisation detector. Mean value of sterols + stanols content was 99.5 mg 100 g$^{-1}$ DW. The main sterol was β-sitosterol. Results showed a year effect on sterol content, whatever the wheat species. This could be explained by the differences in climatic conditions prevailing during plant cycle and grain filling. A large variability for sterols content was found between species and within each species. Emmer wheat revealed the lowest values for all sterols and stanols. Higher values of sterols were obtained in durum wheat. This work is the first report studying *T. carthlicum*, *T. polonicum*, *T. turgidum*, *T. timopheevi*. These species exhibited intermediate values of sterol contents between emmer and durum wheats. Wheat tetraploid species showed interesting levels of sterols and could serve as a great source of these healthy compounds mainly in Mediterranean region where they are consumed as wholegrain. Variation in climatic conditions could help to manage the level of these secondary metabolites.

**Keywords:** phytosterols-phytostanols; metabolic diversity; climatic conditions; durum wheat; emmer wheat; *Triticum carthlicum*; *T. polonicum*; *T. timopheevi*; *T. turgidum*; tetraploid wheats

## 1. Introduction

Among the nutrients contained in cereal grains, phytosterols constitute bioactive secondary metabolites due to beneficial role in the prevention of cardiovascular disease [1–3]. Indeed, these minor lipids, which gained great attention in the last few years, present a similar chemical structure with cholesterol and differed by the presence of a methyl or ethyl group in carbon 24 of the basic cholesterol [4–6]. They have a key-role in the regulation of membrane fluidity and permeability, are involved in embryogenesis, and are precursors of brassinosteroids [7–9]. Sterols occur in plants in different chemical forms and their repartition is dependent on plant species. Sterols are encountered as free sterols, steryl glucosides (SG), and acylated steryl glucosides (ASG), steryl ferulates, steryl esters with phenol acids or fatty acids [5,6,10–12]. The extraction and quantification methods of sterols are well detailed in several reports [5,6,11,13]. Briefly, after lipids extraction, acid hydrolysis is performed, followed by a saponification (alkaline hydrolysis). Acid hydrolysis results in transformation of SG and ASG in free sterols [6]. Quantification methods are numerous using GC of HPLC coupled to FID of MS (the most used methods). More recently, the interest of use of proton nuclear magnetic resonance

(NMR) coupled to partial least square as a fast method to measure squalene and sterols vegetable oils, or liquid chromatography–NMR was reported [14,15].

Phytosterols have been described as bioactive molecules that act to prevent tumor proliferation and cancer formation [2,3,16,17]. Nevertheless, the most studied property of phytosterols is their beneficial impact in the low-density lipoprotein (LDL) cholesterol level reducing and preventing cardiovascular diseases [1,18–22].

Cereals remain the most important source of phytosterols in the human diet based on the daily consumed amounts compared to oilseeds [6,23]. This fact is not surprising, since the most cultivated and consumed seeds around the world are cereals. These compounds are present in two forms in cereals grains: sterols and their saturated forms, stanols. It seems, based on clinical experiments, that phytostanols are more efficient [24]. Phytosterols content in common cereal grains and their by-products have been extensively studied [6,10,23,25–29]. Most of these reports studied sterol content on commercial samples, which considerably limits source traceability. Among the compounds exanimated in the European HEALTHGRAIN project, phytosterols contents in common cereal grains took an important place. A large variability of sterols content was found in *Triticum aestivum*, *T. spelta*, *T. turgidum* var *durum*, *T. turgidum* var *dicoccum*, *T. monococcum* [11,30–32], barley [33], Oat [25,34], rye [26,35]. Several reports have studied seed sterol content of a collection of durum and emmer wheats genotypes [11,30–32,36]. No information is available on the sterol content of other tetraploid wheats. Indeed, these studies have been conducted in Europe. In some of these studies, the growing conditions were not available. Tetraploid wheats are cultivated, mainly, around the Mediterranean Basin and are known for their drought resistance and can be a source for genetic improvement of the durum wheat [37–41].

The objectives of this work were therefore to study the difference in phytosterols and phytostanols content between tetraploid wheat species cultivated under rainfed organic conditions.

## 2. Materials and Methods

### 2.1. Plant Material

Nineteen accessions of different wheat tetraploid species wheat were used in this study. The collection was composed by three accessions of *Triticum turgidum* var. *dicoccum*, *T. turgidum* var *polonicum*, *T. timopheevi*, *T. turgidum* var *carthlicum*, and of *T. turgidum* var. *turgidum* and four durum wheat (*T. turgidum* var. *durum*) (Table 1).

**Table 1.** List of accessions used in this study, their geographical origin.

| Species | Origin |
| --- | --- |
| *Triticum turgidum* var *dicoccum* (Schrank) Thell. | |
| *noricium* Körn. | — |
| *fuschii* Alef. | Spain |
| *seminacum* Krause | — |
| *Triticum turgidum* var *turgidum* L. | |
| *gentile novo* | Portugal |
| *maisani* | Yemen |
| *jodurum* dezassete | Spain |
| *Triticum turgidum* var *polonicum* (L.) MK. | |
| *pseudochrysospermum* | Hungary |
| *skabsubovii* Greb. | Portugal |
| Hadrache | Morocco |
| *Triticum turgidum* var *carthlicum* (Nevski.) MK. | |
| *fuliginosum* n° 48478 | China |
| *fuliginosum* ps 1 | — |

**Table 1.** *Cont.*

| Species | Origin |
|---|---|
| *stramineum* dika 9/14 | Russia |
| *Triticum timopheevi* Zhuk. | |
| *georgia* 29541 | Georgia |
| *typicum* tm4 8340 | — |
| dickson 36.357.1 | Russia |
| *Triticum turgidum* var *durum* (Desf.) MK. | |
| cv. Cham 1 | ICARDA |
| cv. Korifla | ICARDA |
| Gam Goum Rekham | Algeria |
| Jennah Khetifa | Tunisia |

## 2.2. Experimental Conditions

Trials were carried out during two cropping seasons (2010–2011 and 2011–2012) at the Regional Organic Agriculture Experimental Station in Auch (near Toulouse, southwest France, 43°38′47″ N, 0°35′08″ E). The sowings were done on 29 October 2010 and 15 October 2011.

The crops were completely managed under organic and rainfed conditions without any chemical supply. Crushed feathers were applied as an organic fertiliser at a rate of 60 units. ha$^{-1}$ in April 2011 and May 2012. Weeds were mechanically eliminated.

The soil was a clay–loam (organic matter content: 3.2%; pH 8.1) with a depth of about 1.2 m. Table 2 shows temperature and rainfall during the two plant cycles. The growing seasons were characterised by total rainfall of 501 and 483 mm for 2010–2011 and 2011–2012, respectively. These amounts were nearly 14 mm and 32 mm lesser than average rainfall observed for the half-century for the same period in this location (Table 2). Moreover, during grain filling period (from anthesis (Zadoks 60) to full maturity (Zadoks 92) [42] which took place from May to July, both years showed lesser rainfall than the average precipitation of half century (Table 2). Temperatures of both years were nearly 1 °C higher to the temperature observed in the same region during the last 50 years (Table 2). This fact was more marked during the grain filling stage in both years. Indeed, the grain filling period was 2 °C and 6 °C higher in 2011 and 2012, respectively, than the observed average temperature during the 50 past years in this area. It appears, clearly, that the grain filling period coincided with higher temperatures and lower water availability in both years of cultivation.

**Table 2.** Weather conditions during two cropping seasons (2010–2011 and 2011–2012) in Auch (south-west France, near Toulouse). For comparisons, the half-century average values of precipitation and temperatures in the same location were displayed.

| Period | Parameter | 2010–2011 | 2011–2012 | Last 50 Years |
|---|---|---|---|---|
| Grain filling | Rainfall (mm) | 98.5 | 87.9 | 112.9 |
| | Mean temperature (°C) | 18.4 | 22.4 | 16.7 |
| | Rainfall (mm) | 501.2 | 483.4 | 515.0 |
| Plant cycle | Mean temperature (°C) | 12.0 | 12.4 | 11.3 |

Field experiments were conducted as a randomised complete block design with three replicates. Seeds were sown in 12 m$^2$ plots with five rows per plot (20 cm between rows and 3 cm between plants). The anthesis stage took place from beginning to mid-May and from mid-May to end-May for 2012 and 2011, respectively. Full maturity occurred at the end of June for 2012 and the first week of July for 2011.

## 2.3. Seed Moisture Determination

The moisture content was performed by drying the samples, overnight, at 80 ± 2 °C.

*2.4. Sterol–Stanol Determination*

At maturity, grains were harvested in order to determine sterols and stanols content of the grain. The used method for sterol determination is the method performed by Toivo et al. [13] and Alignan et al. [23]. This method permitted the determination of free and esterified sterols. Wheat samples (1.3 g) were put into each tube containing 100 μg of cholestanol (dihydrocholesterol; Aldrich Chemicals Co., Saint-Quentin Fallavier, France) used as an internal standard. The saponification was carried out by adding ethanolic KOH (1 M) (TITRINORM, Prolabo, Pessac, France) and heating for 60 min at 75 °C. Distilled water (1 mL) was added to the samples and the unsaponifiable fraction was extracted from the saponified lipids with 6 mL of *iso*-hexane (Merck, Martillac, France). Sterols and stanols were silylated by *N*-methyl-*N*-trimethylsilyl-heptafluorobutyramide (MSHFBA; Macherey-Nagel, Hoerdt, France) mixed with 1-methyl imidazole (Sigma, Lezennes, France). Sterol and stanol trimethylsilyl ether derivatives (1 μL) were injected into a Perkin-Elmer GC equipped with a CPSIL 5CB 30m column (i. d.: 0.25 mm), and an on-column injector. Detection was performed with a Flame Ionisation Detector (FID). This allows determining free and esterified sterols with fatty acids.

*2.5. Statistical Analyses*

All the data were subjected to variance analysis using the GLM procedure of SAS (SAS Institute, 1987, Cary, NC, USA). The mean pairwise comparisons were based on the means test at 0.5% probability level.

## 3. Results

The collection of tetraploid wheat was studied during two successive years. In this study we determined free and esterified sterols with fatty acids in seeds of wheats. Significant effect of accession was reported on all sterol traits (Table 3). In this studied collection, across the two cropping seasons, mean value of sterols + stanols content was 99.5 mg 100 g$^{-1}$ DW. Sterols represented 78.3% and stanols 21.7% (Table 4). Stanols were composed for one third of campestanol and two thirds of sitostanol. The most abundant sterol is β-sitoterol, which accounted for nearly 70% of total sterols and 55% of sterols + stanols content (Table 4).

**Table 3.** Effect of accession, year and their interaction on sterols and stanols traits measured on grain of 24 accessions of 6 species and subspecies of tetraploid wheats cultivated during two successive seasons (2011 and 2012) in Auch (near Toulouse, southwest of France).

| Trait | Effect | | |
|---|---|---|---|
| | **Accession** | **Year** | **Acc × Year** |
| Campesterol | 17.7 *** | 14.9 *** | 7.9 *** |
| Stigmasterol | 0.9 *** | 0.9 *** | 1.0 *** |
| β-sitoterol | 74.1 *** | 13.8 *** | 50.7 *** |
| δ7-stigmaterol | 0.3 *** | 0.2 ** | 0.2 ** |
| δ7-avenasterol | 0.3 *** | 0.4 *** | 0.4 *** |
| Other sterols | 1.1 *** | 0.9 *** | 0.9 *** |
| Total sterols | 131.2 *** | 2.7 *** | 102.3 *** |
| Campestanol | 8.1 *** | 1.9 ** | 2.4 *** |
| Sitostanol | 21.8 *** | 84.7 *** | 7.8 *** |
| Total stanols | 41.9 *** | 4.9 *** | 0.9 *** |
| Total sterols + stanols | 226.4 *** | 29.7 ** | 174.5 *** |

**, *** significance at 0.01 and 0.001 probability level, respectively.

The impact of year on total sterols as well as individual sterol was significant (Table 3). Nevertheless, the year impact was different depending on species and sterols. Indeed, higher contents of sterols were observed for all species in 2012 except for *T. timopheevi* which recorded a decrease of total sterol–stanol content of 20% in 2012 compared to 2011 (Table 4). Sterols content followed the same

trends in 2012 compared in 2011 (Table 4). In contrast, stanols content presented variable trends. Total stanol and sitostanol contents decreased when temperatures were higher except for *T. turgidum* and *T. polonicum* (Table 4). Campestanol, at the opposite, showed higher values in 2012 than in 2011 except for *T. timopheevi* (Table 4) that exhibited lowest value during 2012. Accession by year interaction was also significant for all measured traits (Table 3).

The studied species differed significantly for all measured traits (Table 3). Total sterol (sterols and stanols, TSS) content varied between 79.3 (Emmer wheat) and 130.0 mg 100 g$^{-1}$ DW (durum wheat). In fact, durum wheat exhibited the highest value and emmer wheat the lowest values whatever the year (Table 4). Similarly, and except for methyl and dimethylsterols (named other sterols in Table 4) emmer wheat exhibited the lowest values for all sterols and stanols, while durum wheat showed the highest ones (Table 4). *T. carthlicum*, *T. polonicum*, *T. turgidum*, *T. timopheevi*, which were not studied up to now presented intermediate mean values. Moreover, *T. polonicum* showed the highest value of stanol content in 2012 (Table 4). In this species as well as durum wheat and *T. timopheevi*, the proportion of stanols represented more than 23.5% of sterol + stanol content. In contrast, emmer wheat, *T. turgidum* and *T. carthlicum*, the contribution of stanols to total sterol + stanol was less than 20%.

A large variation was observed within each species for individual and total sterols (Table 4).

**Table 4.** Free and esterified sterols and stanols contents (in mg 100 g$^{-1}$ DW) and composition of six tetraploid wheat species seeds cultivated wheats cultivated during two successive seasons (2011 and 2012) in Auch (near Toulouse, south-west of France).

| Species | Year | Parameter | Sterol | | | | | | | Stanol | | | Total | Moisture of Seed (%) |
|---|---|---|---|---|---|---|---|---|---|---|---|---|---|---|
| | | | Camp | Stigm | β-Sito | δ7stigm | δ7aven | Other Sterols | Total | Campes | Sitostan | Total | | |
| *Triticum turgidum* var *dicoccum* | 2011 | Mean | 10.2 | 2.3 | 41.3 | 2.9 | 2.9 | 3.7 | 63.4 | 3.6 | 12.2 | 15.8 | 79.3 | 14.5 |
| | | Range | 8.7–12.1 | 1.9–2.4 | 39.7–42.4 | 2.1–3.1 | 2.0–3.2 | 3.1–4.1 | 57.5–67.3 | 3.1–4.0 | 10.1–13.3 | 13.2–17.3 | 70.7–84.6 | 15.0 |
| | 2012 | Mean | 11.7 | 2.6 | 47.5 | 2.8 | 2.8 | 2.2 | 67.3 | 3.8 | 11.5 | 15.3 | 82.6 | 15.0 |
| | | Range | 9.8–12.1 | 2.2–2.8 | 43.9–49.1 | 2.3–3.1 | 2.2–3.0 | 1.7–2.6 | 62.1–72.7 | 3.2–4.0 | 10.5–12.0 | 13.7–16 | 75.8–88.7 | 15.0 |
| *T. turgidum* var *turgidum* | 2011 | Mean | 15.6 | 1.7 | 51.4 | 1.2 | 1.7 | 1.1 | 72.7 | 4.3 | 10.5 | 14.8 | 87.4 | 15.0 |
| | | Range | 13.8–17.2 | 1.2–2.1 | 49.7–52.9 | 0.9–1.4 | 1.5–2 | 0.9–1.2 | 68.0–76.8 | 3.1–4.5 | 10.3–11.1 | 13.4–15.1 | 82.4–91.0 | 14.5 |
| | 2012 | Mean | 17.1 | 3.1 | 57.7 | 1.2 | 1.3 | 1.7 | 80.5 | 6.5 | 12.1 | 18.6 | 99.1 | 14.5 |
| | | Range | 15.6–18.1 | 2.5–3.6 | 53.5–60.2 | 0.9–1.3 | 1.1–1.4 | 1.5–1.8 | 75.1–86.4 | 3.6–6.8 | 11.6–12.5 | 15.1–19.9 | 92.2–106.3 | 14.5 |
| *T. turgidum* var *polonicum* | 2011 | Mean | 16.5 | 3.3 | 45.3 | 0.3 | 1.2 | 0.8 | 68.0 | 8.1 | 15.3 | 23.4 | 91.0 | 15.0 |
| | | Range | 16.1–17.2 | 2.8–3.9 | 43.1–48.2 | 0.2–0.4 | 1.1–1.4 | 0.7–0.9 | 64.1–72.0 | 5.7–8.8 | 14.8–16.3 | 20.5–25.1 | 84.5–97.1 | 15.0 |
| | 2012 | Mean | 19.6 | 4.0 | 66.5 | 1.8 | 1.7 | 2.5 | 93.7 | 9.0 | 17.3 | 28.1 | 121.9 | 14.5 |
| | | Range | 18.2–20.7 | 3.4–4.4 | 63.4–68.2 | 1.6–1.9 | 1.4–1.9 | 2.1–2.6 | 90.1–99.7 | 7.5–9.4 | 17.1–17.8 | 24.6–29.2 | 114.7–126.9 | 14.5 |
| *T. turgidum* var *carthlicum* | 2011 | Mean | 15.3 | 3.0 | 52.1 | 1.5 | 1.3 | 2.9 | 76.3 | 7.0 | 14.0 | 20.9 | 97.2 | 15.0 |
| | | Range | 14.2–15.7 | 2.7–3.3 | 50.1–53.7 | 1.3–1.6 | 1.2–1.4 | 2.6–3.1 | 72.1–78.8 | 5.5–7.7 | 13.3–15.0 | 18.8–22.7 | 90.9–101.5 | 14.5 |
| | 2012 | Mean | 19.5 | 3.4 | 60.8 | 1.4 | 1.1 | 2.4 | 86.2 | 7.7 | 13.0 | 20.6 | 106.8 | 14.5 |
| | | Range | 18.6–20.9 | 3.0–3.7 | 59.8–62.3 | 1.2–1.6 | 0.9–1.2 | 2.1–2.6 | 93.6–92.3 | 4.2–7.9 | 12.4–13.9 | 16.6–21.8 | 100.1–114.1 | 14.5 |
| *T. turgidum* var *durum* | 2011 | Mean | 15.2 | 2.6 | 53.4 | 2.7 | 2.2 | 2.2 | 78.0 | 9.5 | 18.2 | 27.7 | 105.7 | 14.5 |
| | | Range | 14.7–16.5 | 2.2–2.8 | 50.1–55.7 | 2.5–2.9 | 1.4–1.7 | 2.0–2.4 | 72.9–81.9 | 8.7–9.9 | 18.0–19.0 | 26.7–28.8 | 99.8–110.6 | 14.5 |
| | 2012 | Mean | 22.0 | 4.1 | 73.0 | 1.9 | 1.5 | 1.9 | 102.5 | 10.1 | 17.3 | 27.4 | 130.0 | 14.5 |
| | | Range | 20.1–23.2 | 3.9–4.3 | 68.4–74.8 | 1.7–2.1 | 1.3–1.7 | 1.6–2.1 | 97.1–108.2 | 8.6–10.9 | 15.9–18.2 | 24.5–29.1 | 121.5–137.3 | 15.0 |
| *T. timopheevi* | 2011 | Mean | 15.5 | 3.6 | 54.0 | 1.8 | 1.3 | 2.4 | 78.5 | 8.9 | 18.0 | 26.7 | 105.2 | 15.0 |
| | | Range | 14.3–16.9 | 3.1–4.0 | 51.2–56.7 | 1.7–2.0 | 1.1–1.4 | 2.0–2.7 | 73.4–83.7 | 7.7–9.1 | 17.3–18.8 | 25–27.9 | 98.4–111.6 | 15.0 |
| | 2012 | Mean | 13.7 | 2.6 | 49.9 | 1.3 | 0.8 | 1.2 | 68.3 | 6.4 | 13.2 | 19.6 | 87.9 | 14.5 |
| | | Range | 13.0–14.4 | 2.1–3.0 | 47.5–52.6 | 1.0–1.5 | 0.7–0.9 | 1.0–1.3 | 65.3–73.7 | 3.7–6.9 | 12.4–14.6 | 16.1–21.6 | 81.4–95.3 | 14.5 |

Camp: Campesterol; Stigma: stigmasterol; β–Sito: β–Sitosterol; δ7stigma: δ7stigmaterol; δ7aven: δ7avenasterol; Campes: Campestanol; Sitostan: Sitostanol.

## 4. Discussion

The objective of this study was to examine the composition and content of sterols and stanols of tetraploid wheats cultivated in the same location during two years. All species were cultivated under the same growing conditions, and therefore the observed differences between tetraploid wheat were due to genetic factors. A broad range of variation for sterols–stanols content was observed between tetraploid species (Tables 3 and 4). The result was expected, according to the scarce reports published on sterols content in wheats species. Indeed, Nurmi et al. [30], within the European framework HEALTHGRAIN, showed a large range of sterols in hexaploid (winter and spring wheats, spelt), tetraploid (durum and emmer), and diploid (eikorn) wheat. Moreover, it has been reported that emmer and durum wheats contain higher sterol levels than hexaploid ones [11,32,36,43]. There is no information on sterol content in other tetraploid wheats. This is the first report on *Triticum turgidum*, *T. polonicum*, *T. carthlicum,* and *T. timopheevi*. The mean value of sterol+stanol content, across the years, was 81.0 and 117.9 mg. 100 g$^{-1}$ DW for emmer and durum wheats, respectively. The observed value for emmer was lower than those observed by Nurmi et al. [30]. In contrast, durum wheat exhibited higher values than those showed in the HEALTHGRAIN network, and those reported in Italy [11,32,36]. These differences could be explained by both environmental and genotypic effects [23,25,26,43–46]. Indeed, the studied accessions within each species were different between the both reports. Moreover, the climatic conditions probably influenced the sterol content, reported here and in Nurmi et al. [30]. In addition, our experiments were carried out under organic cultivation that could be considered as stressed growing conditions which could influence the potential expression of the genotypes [23]. Otherwise, all these species showed higher sterols + stanols content than bread wheat [23,30]. Moreover, different methods of extraction and quantification of sterols were used in all these studies including our work. Indeed, in both HEALTHGRAIN network and Italian studies, extraction involved an acid hydrolysis before saponification, while in our method, alkaline hydrolysis was performed with acid one. In addition, Caboni et al. have used GC–MS for sterols quantification and identification. In contrast, we quantify sterols by GC–FID as a frequently used technique [5,6]. This is confirmed by the studies of Caboni team [11,32,36]. In another hand, some of these works did not present the origin of wheat grains and the plant growth conditions. Alignan et al. [23] considered that plant growth under organic cultivation as stressed conditions. Furthermore, it has been reported that conditions and duration of storage could influence the sterol composition [47]. All these reasons could participate in the observed differences in this study compared to those already performed in emmer and durum wheat [11,30,32,36].

Climatic conditions during the two cropping seasons of this study impacted the sterol–stanol content in tetraploid wheat species (Tables 3 and 4). Indeed, 2012 was hotter and less rainy than 2011, especially during grain filling period (Table 2). Nevertheless, species showed different reaction to these conditions. Higher temperatures and lower rain recorded in 2012 resulted in higher sterol + stanol content except for *T. timopheevi* which exhibited a reduction of this trait under such conditions (Table 4).

Weather conditions, high temperatures, and drought have been reported to influence sterol content by modifying the activity of enzymes involved in sterol biosynthesis pathway [9,44,45,48]. Indeed, in these reports, it was observed that higher temperatures or drought increased sterols content due to the augmentation of the activity of 3–hydroxy–3–methylglutaryl coenzyme A reductase, called HMG–CoA reductase [45], which is a key enzyme in sterols biosynthesis. This fact was supported by the increase of sterol content and therefore total sterol content (Table 4).

The different behavior noticed for *T. timopheevi* compared to the other tetraploid wheat species could be explained by genetic factors and cultivation areas. Indeed, it is well known that *T. timopheevi* has GGAA genome, unlike the other five species that belong to the same BBAA genome group. Moreover, the domesticated form of this species is cultivated only in western Georgia [49]. The region of domestication is much colder than the Fertile Crescent. Within the BBAA genome species, two groups can be depicted according to their sterol profiles and their behaviour to changing weather conditions. The first one composed by *T. carthlicum*, durum, and emmer wheats which presented an increase in

sterols content resulting in a rise of sterol + stanol content. The second one, constituted by *T. tugidum* and *T. polonicum* that higher total sterol in 2012 was induced by an increase in both sterol and stanol contents (Table 4). The increase in sterol + stanol content was induced by the rise of the major sterol: β–sitosterol and its saturated form sitostanol (Table 4). The augmentation of this individual sterol may result from the intensification of overall enzymes of biosynthesis pathway and in particular sterol–methyl–transferase 2 (SMT2) [9,48,50,51]. SMT2 allows to production of β stitosterol and stigmasterol instead of campesterol and its activity is dependent on temperature during grain filling. It was not the case in our study. Indeed, for the majority of wheat species, higher temperatures in 2012 induced an increase of β sitosterol and total sterols compared to 2011 without a decrease of campesterol (Table 4). This fact allows confirming that the climatic conditions prevailing during 2011 and 2012 have probably impacted the total sterol content by influencing the activity of enzymes involved in earlier stages of biosynthesis sterol pathway [4,44,45,48,50].

Sterol and stanol content in bread wheat have been reported to not interfere with protein content [23]. Therefore, these traits could be taken into account in breeding programs without any interference with protein content, an important technological character considered in breeding programs.

## 5. Conclusions

Wheats are the major sources of sterols and stanols when consumed as wholegrain. A wide variability of sterol and stanol content exists between studied tetraploid wheat species beyond durum and emmer wheat species. Interesting level of sterols is present in seeds of these species and could be used as source for human alimentation consumed as wholegrain as in the Mediterranean areas. Durum wheat, which is broadly used in Mediterranean countries, contains the highest levels of sterols and stanols. At the opposite, emmer wheat presented the lowest level of sterols in the seeds. Although the accession impact remains important, our results showed that climatic conditions significantly influenced the sterols and stanols content in these species. Nevertheless, it is important to ascertain these results by investigating more accessions in further environments and climatic conditions. It can be suggested that the impact of weather conditions, in climate change context, could be exploited easily in order to stimulate sterols content in cereal seed and to offer an added value to the tetraploid wheat species.

**Author Contributions:** O.M. conceptualization. Z.M. and O.M developed the methodology and performed the experiments and the measurements. O.M. assisted with measurements. O.M. and Z.M. contributed to the analysis and interpretation of the data and to the writing of the manuscript.

**Funding:** This study was performed with the laboratory's own funds.

**Acknowledgments:** Authors are grateful to Muriel Cerny, from the Laboratory of Agroindutrial Chemistry, for the technical help in carrying on sterol measurements. Authors thank Philippe Monneveux for providing seeds and the critical review of the manuscript.

**Conflicts of Interest:** The authors declare no conflict of interest.

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
