# Peer review of "Tetraploid Wheats: Valuable Source of Phytosterols and Phytostanols"

_agronomy, doi:10.3390/agronomy9040201_

Round 1
Reviewer 1 Report
Manuscript "Tetraploid wheats: valuable source of phytosterols and phytostanols" provides data on phytosterols and phytostanol content in selected accessions representing varieties of Triticum turgidum and T.timopheevi. Overall, research is well justified and important for human health.
Taxonomical classification should be constantly used across manuscript (Table 1). Carbon isotope (line 64) discrimination? - where are results – what is purpose? Materials should be supplemented with accession number in genebank (availability of plant materials) – otherwise results will be of low interest. Data on seed moisture is missing and should be supplemented in methods.
Different varieties can be represented by different accessions and in case of reduced representation it is hard to get reliable results. Similarly influence of environment is not very representatively studied. Level of expression of some genes involved in biosynthesis of sterols should be possibly assessed during grain filling. Manuscript needs thorough editing.
I would question "genetic diversity" - this is more "metabolic diversity" resulting from genotype, phenotype and interaction. Conclusions (lines 208-209) is not so straightforward - as two seasons are not enough to reckon about influence of environment (about 12 environments would be better in case of quantitative traits) -there can also be conflict with accumulation of increase of soluble sugars that are precursors in phytosterol synthesis pathway and they may result from sprouting and may be attractive for fungal diseases.
Author Response
Dear Reviewer,
Thank you for your valuable comments.
All modifications have been reported in blue in the manuscript. This manuscript aimed an evaluation of sterols in tetraploid wheat species from agronomic view.
Manuscript "Tetraploid wheats: valuable source of phytosterols and phytostanols" provides data on phytosterols and phytostanol content in selected accessions representing varieties of Triticum turgidum and T.timopheevi. Overall, research is well justified and important for human health.
Taxonomical classification should be constantly used across manuscript (Table 1). Carbon isotope (line 64) discrimination? - where are results – what is purpose?
Answer: This is a mistake. The title of Table 1 was corrected
Materials should be supplemented with accession number in genebank (availability of plant materials) – otherwise results will be of low interest.
Answer: Seeds were provided by ICARDA. Information is available in ICARDA Archives. Nevertheless, presently we cannot access to this information du to headquarter modification.
Data on seed moisture is missing and should be supplemented in methods.
Answer: This information was added in table 4.
Different varieties can be represented by different accessions and in case of reduced representation it is hard to get reliable results.
Answer: This point is questionable. This manuscript aimed to present differences between several species of tetraploid wheats. Some of these species are not much cultivated. It is sometimes difficult to have significant representability. However, all studied species are represented with the same number of accession except for durum wheat which is represented by four accessions. This seems to us quite balanced.
Similarly influence of environment is not very representatively studied. Level of expression of some genes involved in biosynthesis of sterols should be possibly assessed during grain filling. Manuscript needs thorough editing.
Answer: This manuscript aimed to compare sterol and stanol content in seed of different tetraploid wheat species. Expression of genes involved in biosynthesis was not studied here. Nevertheless, an interesting recent work of Kumar et al. (2018) was considered in this manuscript (Kumar et al., Regulation of phytosterol biosynthetic pathway during drought stress in rice. Plant Physiol. Biochem. 2018, 129, 11–20.)
I would question "genetic diversity" - this is more "metabolic diversity" resulting from genotype, phenotype and interaction.
Answer: Thank you for your comment. The observed results showed the variation of seed sterol content of tetraploid wheats cultivated in the same location. We changed genetic diversity by metabolic diversity.
Conclusions (lines 208-209) is not so straightforward - as two seasons are not enough to reckon about influence of environment (about 12 environments would be better in case of quantitative traits) -there can also be conflict with accumulation of increase of soluble sugars that are precursors in phytosterol synthesis pathway and they may result from sprouting and may be attractive for fungal diseases.
Answer: This study aimed to investigate the content of sterols and stanols in some tetraploid wheats. Up to now, the reports were done on seed purchased in the market or on one year experiment. Few ones have studied interactions genotype x environment on three year experiments in Hungary (Nurmi et al., 2010). Some others studied effect of drought, under controlled conditions, on sterol biosynthesis pathway. This study is a first report one on some tetraploid wheat species. It is a first step that opens other perspectives and other ways for further and deeper investigations. This seems to us quite balanced. However, we nuanced this point in conclusion.
English was reviewed by an English native.
The presentation of M&M, reseach design and conclusion was improved in the revised version of the maunuscript
best regards
Reviewer 2 Report
The paper “Tetraploid wheats: valuable source of phytosterols and phytostanols” examined phytosterol and phytostanol content in seed of six tetraploid wheat 12 species cultivated during two successive years under rain fed organic conditions in France. The paper suggests as cereal could be an important source of phytosterols, but is necessary underline that is true only consuming the wholegrain.
The paper present some important criticism on analytical method that are due to a poor bibliographic evaluation which, in fact, is very lacking. In cereals, phytosterols occur as free sterols, steryl esters with fatty acids, or phenolic acids, steryl glycosides, and acylated steryl glycosides (Piironen, V. et al. ; Plant sterols: Biosynthesis, biological function and their importance to human nutrition. J. Sci. Food Agric. 2000, 80, 939-946. Morrison, W. R. Cereal lipids. In Advances in Cereal Science and Technology; Pomeranz, Y., Ed.; Am. Assoc. Cereal Chem. Inc.: St. Paul, MN, 1978; Vol. II, pp 221-348. Maatta, K. Et al.; Phytosterol content in seven oat cultivars grown at three locations in Sweden. J. Sci. Food Agric. 1999, 79, 1021-1027.) moreover, to collect total sterol is necessary to provide also a hydrolysis under acidic conditions (Piironen, V.; Toivo, J.; Lampi, A.-M. Plant sterols and cereal products. Cereal Chem. 2001, 79, 148-154), where in the present paper the authors carried out a saponification that allows the collection of free sterols and sterols esterified with fatty acids (Iafelice G., et al. "Characterization of total, free and esterified phytosterols in tetraploid and hexaploid wheats",2009,"Journal of Agricultural and Food Chemistry", 57, 2267-2273). In fact, bound sterols, mainly in tetraploid wheat represent at least 20% of true total sterols. The authors reported that are lacking data on tetraploid wheats, but I suggest to read: Pelillo M., et al.,"Identification of plant sterols in hexaploid and tetraploid wheats using gas chromatography with mass spectrometry",2003,"Rapid Communications in Mass Spectrometry", 17, 2245-2252 and Caboni M.F. et al. "Analysis of fatty acid steryl esters in tetraploid and hexaploid wheats: Identification and comparison between chromatographic methods", 2005,"Journal of Agricultural and Food Chemistry",53, 7465-7472.No information where done on identification method (literature or standard retention time comparation?)
Author Response
Dear Reviewer,
Thank you so much for the precisous remarks.
The paper “Tetraploid wheats: valuable source of phytosterols and phytostanols” examined phytosterol and phytostanol content in seed of six tetraploid wheat 12 species cultivated during two successive years under rain fed organic conditions in France.
The paper suggests as cereal could be an important source of phytosterols, but is necessary underline that is true only consuming the wholegrain.
Answer: This was added (Abstract L23; Conclusion P9 L209-210)
The paper present some important criticism on analytical method that are due to a poor bibliographic evaluation which, in fact, is very lacking. In cereals, phytosterols occur as free sterols, steryl esters with fatty acids, or phenolic acids, steryl glycosides, and acylated steryl glycosides
(Piironen, V. et al. ; Plant sterols: Biosynthesis, biological function and their importance to human nutrition. J. Sci. Food Agric. 2000, 80, 939-946.
Morrison, W. R. Cereal lipids. In Advances in Cereal Science and Technology; Pomeranz, Y., Ed.; Am. Assoc. Cereal Chem. Inc.: St. Paul, MN, 1978; Vol. II, pp 221-348.
Maatta, K. Et al.; Phytosterol content in seven oat cultivars grown at three locations in Sweden. J. Sci. Food Agric. 1999, 79, 1021-1027.)
moreover, to collect total sterol is necessary to provide also a hydrolysis under acidic conditions (Piironen, V.; Toivo, J.; Lampi, A.-M. Plant sterols and cereal products. Cereal Chem. 2001, 79, 148-154), where in the present paper the authors carried out a saponification that allows the collection of free sterols and sterols esterified with fatty acids
(Iafelice G., et al. "Characterization of total, free and esterified phytosterols in tetraploid and hexaploid wheats",2009,"Journal of Agricultural and Food Chemistry", 57, 2267-2273).
Answer: Thank you for this comment. These publications were judicious and helped in the discussion. Information concerning extraction and quantification of sterols was added in introduction (P1 L35-45) and discussed P8 L191-200.
In fact, bound sterols, mainly in tetraploid wheat represent at least 20% of true total sterols. The authors reported that are lacking data on tetraploid wheats, but I suggest to read:
Pelillo M., et al.,"Identification of plant sterols in hexaploid and tetraploid wheats using gas chromatography with mass spectrometry",2003,"Rapid Communications in Mass Spectrometry", 17, 2245-2252 and
Caboni M.F. et al. "Analysis of fatty acid steryl esters in tetraploid and hexaploid wheats: Identification and comparison between chromatographic methods", 2005,"Journal of Agricultural and Food Chemistry",53, 7465-7472.
Answer: Thank you for this comment. These publications were helpful in introduction (P2 L54-61) and discussion (P8 L175-177, 181, and 188-197) sections. Indeed, we made a distinction between what was already published (emmer and durum wheat) and the novelty highlighted in our study.
No information where done on identification method (literature or standard retention time comparation?)
Answer: This was discussed P8 L188-197).
The manuscript was improved according the remarks.
Reviewer 3 Report
This work performs analysis and the comparison of phytosterols and phytostanols amounts of different wheat species under different cultivated conditions. The idea is not extremely original and the work is not very complex. The manuscript fits within the scope of the journal. However, some critical lacks have been detected and should be solved prior to accept the manuscript:
· Taking into account that the aim of this research is to study the amounts of phytosterols and phytostanols as a function of the genotypic variability of wheat, a validation of the GC method should be provided in terms of linearity, limit of quantification (LOQ), limit of detection (LOD), accuracy and precision, according to the document SANCO/12571/2013.
· Line 201. The authors should be careful about in their conclusions. Please, see Shi et al 2019 Food Chemistry 287, 46-54 (https://doi.org/10.1016/j.foodchem.2019.02.072).
Author Response
Dear reviewer,
Many thanks for your valuable remarks.
The manuscript was reviewed according your comments.
This work performs analysis and the comparison of phytosterols and phytostanols amounts of different wheat species under different cultivated conditions. The idea is not extremely original and the work is not very complex. The manuscript fits within the scope of the journal. However, some critical lacks have been detected and should be solved prior to accept the manuscript:
· Taking into account that the aim of this research is to study the amounts of phytosterols and phytostanols as a function of the genotypic variability of wheat, a validation of the GC method should be provided in terms of linearity, limit of quantification (LOQ), limit of detection (LOD), accuracy and precision, according to the document SANCO/12571/2013.
Answer: The method performed in this study was already used in several studies around the world and results were published some of them are listed below and cited in the manuscript. Nevertheless, the used method was carried out in order to screen large number of samples mostly in agronomy and genetics. Cabon et al. ("Analysis of fatty acid steryl esters in tetraploid and hexaploid wheats: Identification and comparison between chromatographic methods", 2005,"Journal of Agricultural and Food Chemistry",53, 7465-7472.) showed that FASE (GC or HPLC) method are useful to discriminate plant materials). New analytical methods have been developed in chemistry and food sciences. The differences between our results and those reported in recent studies are presented in discussion section.
Pelillo M., et al.,"Identification of plant sterols in hexaploid and tetraploid wheats using gas chromatography with mass spectrometry",2003,"Rapid Communications in Mass Spectrometry", 17, 2245-2252.
Caboni M.F. et al. "Analysis of fatty acid steryl esters in tetraploid and hexaploid wheats: Identification and comparison between chromatographic methods", 2005,"Journal of Agricultural and Food Chemistry",53, 7465-7472.
Iafelice G., et al. "Characterization of total, free and esterified phytosterols in tetraploid and hexaploid wheats",2009,"Journal of Agricultural and Food Chemistry", 57, 2267-2273
Alignan, et al. Effects of genotype and sowing date on phytostanol–phytosterol content and agronomic traits in wheat under organic agriculture. Food Chem. 2009, 117(2), 219–225. https://doi.org/10.1016/j.foodchem.2009.03.102
Harrabi, et al. , S. Phytostanols and phytosterols distributions in corn kernel. Food Chem. 2008, 111, 115–120. doi:10.1016/j.foodchem.2008.03.044
Merah, et al. Genetic control of phytosterol content in sunflower seeds. Theor. Appl. Genet. 2012, 125, 1589–1601. doi: 10.1007/s00122-012-1937-0.
Roche, et al. Sterol concentration and distribution in sunflower seeds (Helianthus annuus L.) during seed development. Food Chem. 2010, 119, 1451-1456. https://doi.org/10.1016/j.foodchem.2009.09.026
Roche, et al. Sterol content in sunflower seeds (Helianthus annuus L.) as affected by genotypes and environmental conditions. Food Chem. 2010, 121, 990-995. https://doi.org/10.1016/j.foodchem.2010.01.036
Roche, et al. Fatty acid and phytosterol accumulation during seed ripening in three oilseed species. Internat. J. Food Sci. Technol. 2016, 51, 1820–1826. https://doi.org/10.1111/ijfs.13153
· Line 201. The authors should be careful about in their conclusions.
Please, see Shi et al 2019 Food Chemistry 287, 46-54 (https://doi.org/10.1016/j.foodchem.2019.02.072).
Answer: Thank you for this comment. This publication was helpful in introduction and discussion sections.
Round 2
Reviewer 1 Report
I generally accept all explanations and corrections. Conclusions were supplemented.
Reviewer 3 Report
In my judgment, after review carefully the manuscript, the changes are adequate and the authors have improved it properly. The confusing issues are also clearer. Currently, the article includes all the necessary information for a proper understanding of the work and results are of interest.